# Treatment strategies for stage IA non-small cell lung cancer: A SEER-based population study

**Bo Wu[1]ʘ, Xiang Zhang[1]ʘ, Nan Feng[1], Zhuozheng Hu[1], Jiajun Wu[1], Weijun Zhou[1], Yiping Wei[1], Wenxiong Zhang[1]‡\*, Kang Wang[2]‡\***

**1** Department of Thoracic Surgery, The Second Affiliated Hospital, Jiangxi Medical College, Nanchang University, Nanchang, 330006, China, **2** Department of Traditional Chinese Medicine, The Second Affiliated Hospital, Jiangxi Medical College, Nanchang University, Nanchang, 330006, China

ʘ These authors contributed equally to this work.
‡ These authors also contributed equally to this work.
\* zwx123dr@126.com (WZ); wangkangmissyou@163.com (KW)

**Data Availability Statement:** All relevant data are within the paper and its Supporting Information files.

**Funding:** This study was funded by the National Natural Science Foundation of China 81560345, to

## Abstract

### Background

There are various therapeutic methods for treating stage IA (T1N0M0) non-small cell lung cancer (NSCLC), but no studies have systematically assessed multiple treatments to determine the most effective therapy.

### Methods

Stage IA NSCLC patient data collected between 2004 and 2018 were gathered from the Surveillance, Epidemiology, and End Results (SEER) database. Treatment modalities included observation, chemotherapy alone (CA), radiation alone (RA), radiation+chemotherapy (RC), surgery alone (SA), surgery+chemotherapy (SC), surgery+radiation (SR) and surgery+radiation+chemotherapy (SRC). Comparisons were made of overall survival (OS) and lung cancer-specific survival (LCSS) among patients based on different therapeutic methods by survival analysis.

### Results

Ultimately, 89147 patients with stage IA NSCLC between 2004 and 2018 were enrolled in this study. The order of multiple treatment modalities based on the hazard ratio (HR) for OS for the entire cohort revealed the following results: SA (HR: 0.20), SC (HR: 0.25), SR (HR: 0.42), SRC (HR: 0.46), RA (HR: 0.56), RC (HR: 0.72), CA (HR: 0.91) (P<0.001), and observation (HR: Ref). The SA group had the best OS and LCSS, and similar results were found in most subgroup analyses (all P<0.001). The order of surgical modalities based on the HR for OS for the entire cohort revealed the following results: lobectomy (HR: 0.32), segmentectomy (HR: 0.41), wedge resection (HR: 0.52) and local tumor destruction (HR: Ref). Lobectomy had the best effects on OS and LCSS, and similar results were found in all subgroup analyses (all P<0.001).

WZ and Natural Science Foundation of Jiangxi Province, 20212BAB206050, to WZ. The funders had no role in study design, data collection and analysis, decision to publish, or preparation of the manuscript.

**Competing interests:** The authors have declared that no competing interests exist.

**Abbreviations:** AJCC, The American Joint Committee on Cancer; CA, Chemotherapy alone; CI, Confidence interval; HR, Hazard ratio; LADC, Lung adenocarcinoma; LC, Lung cancer; LCSS, Lung cancer-specific survival; LSCC, Lung squamous cell cancer; NSCLC, Non-small-cell lung cancer; OS, Overall surviva; RA, Radiation alone; RC, Radiation+Chemotherapy; SA, Surgery alone; SBRT, Stereotactic body radiotherapy; SC, Surgery +Chemotherapy; SEER, The Surveillance, Epidemiology and End Results; SR, Surgery +Radiation; SRC, Surgery+Radiation +Chemotherapy.

## Conclusion

**SA** appeared to be the optimal treatment modality for **patients** with stage IA NSCLC, and lobectomy was associated with the best prognosis. There may be some indication and selection bias in our study, and the results of this study should be confirmed in a prospective study.

## Introduction

Lung cancer (LC) is the most common cause of cancer-related morbidity and mortality globally, and 238,340 patients were newly diagnosed with LC in the United States in 2022 [1]. Approximately 85% of LC cases are non-small cell lung cancer (NSCLC), and the incidence rate of NSCLC is increasing every year worldwide [2]. Although most early-stage LC patients are not easily diagnosed based on clinical manifestations, the application of low-dose lung computed tomography to conventional LC screening may increase the detection rate of early-stage LC [3]. In addition to traditional treatment modalities, some recent studies have confirmed the correlation between various treatment interventions for stage IA NSCLC and the improvement of prognosis [4,5]. Currently, the treatment modalities for stage I NSCLC include surgery, radiotherapy, chemotherapy, laser ablation, cryosurgery, electrocautery, fulguration and combined treatments. Moreover, studies have shown that for patients with localized NSCLC (stages I to III), the addition of immunotherapy to conventional curative surgery or radiation therapy provides a survival advantage [6]. Therefore, the multiple treatment choices need to be systematically evaluated and compared to determine the most effective treatment modality.

A meta-analysis showed that stereotactic body radiotherapy (SBRT) has the potential to be an alternative to surgery for patients with stage I/II NSCLC [7]. Some studies have shown that surgery combined with chemotherapy may be a viable treatment option [8]. Other studies have shown that radical radiotherapy is one of the therapeutic tools for stage I NSCLC [9]. Thus, the best treatment modality option is still controversial. Regarding the choice of surgical modality, a study published in the 1960s showed a better prognosis for lobectomy than limited resection for IA NSCLC [10]. However, recently, many meta-analyses have suggested that segmentectomy may be preferable to lobectomy in patients with stage IA NSCLC [11]. Some studies have indicated that prognosis following wedge resection appears to be better than that after segmentectomy and lobectomy [12]. Recently, some scholars have suggested that there appears to be a relative benefit of segmentectomy compared to wedge resection because of its superiority in regional control [13]. Thus, the multiple potential surgical modalities also need to be systematically evaluated to determine the best surgical treatment.

Therefore, we sought to address the optimal treatment modalities for stage IA NSCLC patients by comparing overall survival (OS) and lung cancer-specific survival (LCSS) among different treatment methods based on the Surveillance, Epidemiology, and End Results (SEER) database.

## Methods

### Patient and public involvement

The SEER database provides information on cancer statistics free of charge, so this study did not require a patient and public involvement statement.

## Data source

This study was carried out according to the Declaration of Helsinki guidelines [14]. This was a retrospective observational population-based epidemiological study of survival in stage IA NSCLC patients who underwent different treatments, and the data obtained in 2022 were lung cancer data updated between 2004 and 2018 from the SEER database (https://seer.cancer.gov/). The details of IA NSCLC were retrieved using SEER * stat version 8.3.9. Tumor staging was performed using the 8th edition TNM staging system of the American Joint Committee on Cancer (AJCC) [15]. Tumor histology was coded using the ICD-0-3/WHO 2008.

The inclusion criteria included the following: (I) primary site of C34-Lung; (II) histologically confirmed NSCLC; (III) IA stage (T1N0M0); and (Ⅳ) year of diagnosis from 2004–2018. The exclusion criteria were as follows: (I) unknown survival months, (II) unknown cancer-specific survival, (III) unknown specific treatment method, and (Ⅳ) diagnosis of NSCLC after death (including an autopsy or death certificate).

## Construction of variables

We collected 11 demographic and clinical patient variables. Patient characteristics included age, sex, year of diagnosis, ethnicity, histological type, primary location, surgery, chemotherapy, radiation, marital status and treatment modality. In this study, surgical modalities were classified into local tumor destruction (including laser ablation, cryosurgery, electrocautery and fulguration), wedge resection, segmentectomy and lobectomy. The years of diagnosis included 2004–2008, 2009–2013 and 2014–2018. To better analyze the survival outcomes of the different treatment modalities, we added a variable called treatment modality. Treatment modalities were categorized as observation, chemotherapy alone (CA), radiation alone (RA), radiation+chemotherapy (RC), surgery alone (SA), surgery+chemotherapy (SC), surgery+radiation (SR) and surgery+radiation+chemotherapy (SRC). Thus, 8 variables (age, sex, year of diagnosis, ethnicity, histological type, primary location, marital status and treatment modality) were included in the prognostic analysis. OS and LCSS were the primary outcomes in this study.

## Statistical analysis

Categorical variables were tabulated according to frequency and percentage, and comparative prognostic factors of OS and LCSS were assessed via univariate and multivariate Cox proportional hazard models. Subgroup analyses were performed for categorical variables with P<0.01 in the univariate Cox analysis. Thus, the IA NSCLC patients were divided into different subgroups: age (≤65 or >65 years), sex (male or female), ethnicity (black or white or Asian or Pacific Islander or American Indian/Alaska Native), years of diagnosis (2004–2008 years or 2009–2013 years or 2014–2018 years), histological type [lung adenocarcinoma cell cancer (LADC) or lung squamous cell cancer (LSCC) or others], location (upper lobe or middle lobe or lower lobe) and marital status (married or single). Kaplan–Meier survival curves were used to compare survival among different treatment modalities in different subgroups. The hazard ratio (HR) and 95% confidence interval (CI) were calculated for the Cox proportional hazard models. All statistical calculations were completed with the R (version 4.1.3) software package (**https://www.r-project.org/**). The results were considered statistically significant at a P value<0.05.

## Results

### Basic characteristics

A total of 95767 stage IA LC patients whose data were collected between 2004 and 2018 were identified in the SEER database. After applying the inclusion and exclusion criteria, we

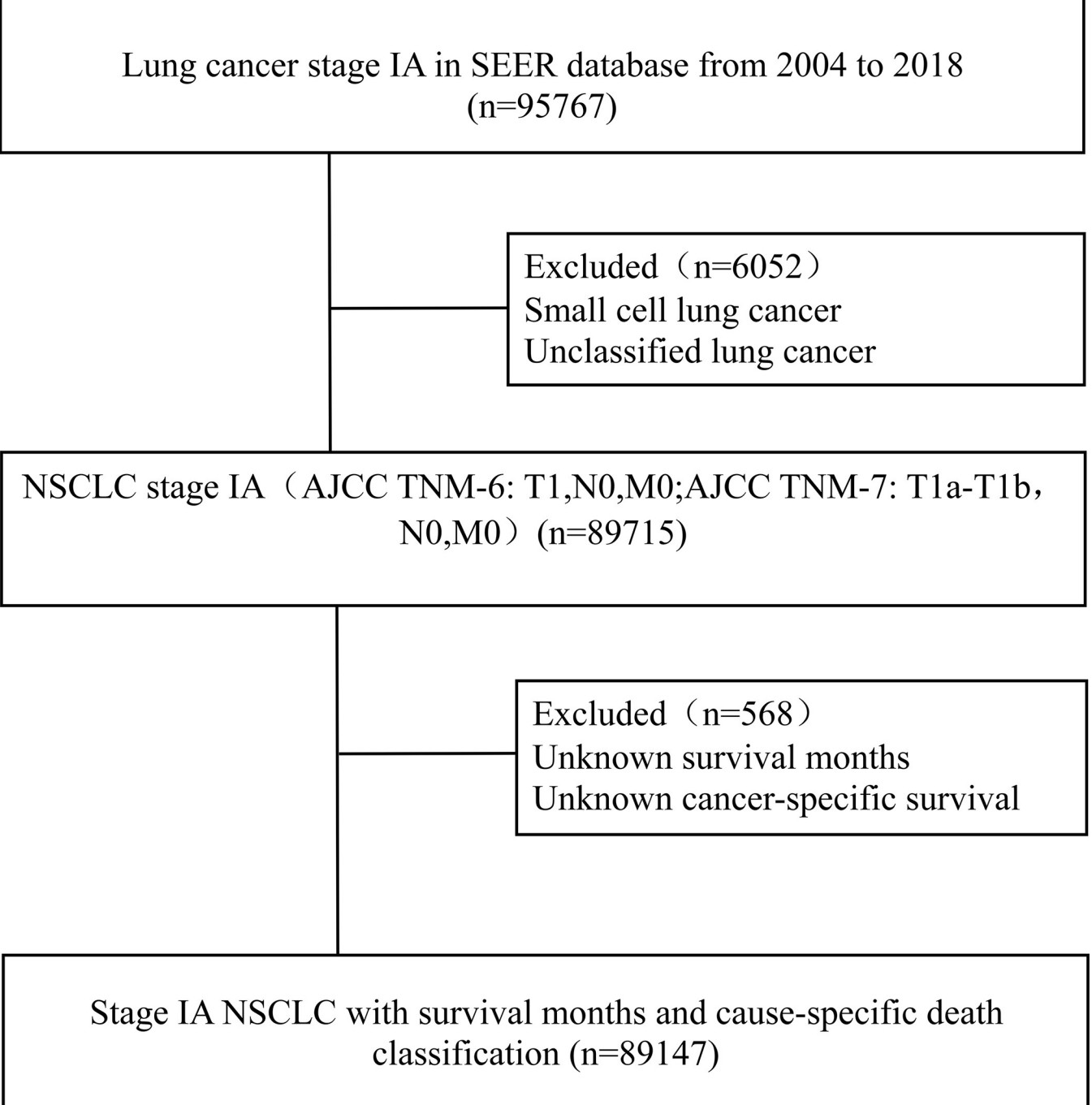

**Fig 1. Flow diagram of enrollment.**

ultimately enrolled 89147 stage IA NSCLC patients with survival months and lung cause-specific death classification. The flow chart of the patient selection process is shown in **Fig 1.** Among the 89147 patients, the median age was 71 years, 55.67% were female, and 43.04% were single. In the case of those still alive, 56.92% of patients were followed up for more than five years. There was no statistically significant difference between missing and nonmissing data

(P>0.05), and the missing data were deleted. The baseline characteristics of all patients are reported in **Table 1**.

The multiple treatment modalities were as follows: observation (8055, 9.04%), CA (1,058, 1.19%), RA (19,688, 22.08%), RC (1762, 1.98%), SA (55262, 61.99%), SC (1804, 2.02%), SR (1074, 1.20%) and SRC (444, 0.50%). RA included beam radiation, a combination of beam radiation with implants or isotopes, radioactive implants and radioisotopes. SBRT is one form of beam radiation. There was a possibility of indication and selection bias associated with the included populations receiving different treatments. In addition, subgroup analyses could not be performed because not all treatments were performed in American Indian/Alaska Native patients. The baseline characteristics of the stage IA NSCLC patients treated by multiple treatment modalities are reported in **Table 2**.

## Univariate and multivariate Cox analyses

Comparative prognostic factors of OS and LCSS were evaluated by univariate and multivariate Cox proportional hazard models. The analyses revealed that, excluding primary location, age>65, male sex, American Indian/Alaska Native ethnicity, years of diagnosis from 2004–2008, LSCC and single status were prognostic factors for poor OS and LCSS. Moreover, treatment modality was an independent prognostic factor for OS and LCSS. These results held true after adjusting for other confounders in the multivariate Cox analysis. In **Table 3**, patients treated with SA had the best survival outcome. The adjusted HRs of OS for observation, CA, RA, RC, SA, SC, SR and SRC were 0.94, 0.57, 0.70, 0.23, 0.29, 0.41 and 0.51, respectively (P<0.001). As summarized in **Table 4**, patients who underwent lobectomy had the best survival outcome. The adjusted HRs of OS for local tumor destruction, wedge resection, segmentectomy and lobectomy were 0.59, 0.49 and 0.39, respectively (P<0.001).

## Comparison of OS and LCSS among multiple treatment modalities

The overall HR for OS of the entire cohort based on multiple treatment modalities increased as follows: SA, SC, SR, SRC, RA, RC, CA and observation (**Fig 2A and 2C**). However, in the age ≤65 years and married subgroups, the positions of CA and observation were interchanged (**Fig 3A and 3C, S1A and S1C Fig** in S1 File). In the diagnosed 2014–2018, other histology and middle lobe subgroups, the order of modalities based on an increasing HR was as follows: SA, SC, SR, RA, SRC, RC, CA and observation (**S2-S4A and S4C Figs** in S1 File). In the black ethnicity subgroup, the positions of SRC and SR were interchanged (**S5A and S5C Fig** in S1 File). Moreover, in the LSCC subgroup, the order of modalities based on an increasing HR was as follows: SC, SA, SRC, SR, RA, RC, CA and observation (**S6A and S6C Fig** in S1 File). Except for the subgroups mentioned above, the order of modalities based on an increasing HR was the same as the overall trend (**S7-S17A and S17C Figs** in S1 File).

The overall HR for LCSS of the entire cohort among multiple treatment modalities increased as follows: SA, SC, SR, RA, SRC, RC, observation, and CA (**Fig 2B and 2D**). However, in the age ≤65 years subgroup, the positions of RC and observation were interchanged (**Fig 3B and 3D**). In the other histology subgroups, the order of modalities based on an increasing HR was as follows: SA, SC, SRC, SR, RC, RA, CA, and observation (**S3B and S3D Fig** in S1 File). In the middle lobe subgroup, the order of modalities based on an increasing HR was as follows: SA, SR, SC, RA, SRC, RC, observation, and CA (**S4B and S4D Fig** in S1 File). In the black ethnicity and LSCC subgroups, the order of modalities based on an increasing HR was as follows: SA, SC, SRC, SR, RA, RC, CA, and observation (**S5-S6B and S6D Figs** in S1 File). In the Asian or Pacific Islander ethnicity subgroup, the order of modalities based on an increasing HR was as follows: SA, SC, SR, SRC, RA, observation, RC, and CA

**Table 1. Characteristics of study patients with stage IA NSCLC.**

| Variable | Case (%) |
|---|---|
| **Total** | **89147** |
| **Age** | |
| ≤65 | 23330 (26.17) |
| >65 | 65817 (73.83) |
| **Sex** | |
| Female | 49624 (55.67) |
| Male | 39523 (44.33) |
| **Ethnicity** | |
| White | 76056 (85.32) |
| Black | 7739 (8.68) |
| Asian or Pacific Islander | 4721 (5.30) |
| American Indian/Alaska Native | 405 (0.45) |
| Unknown | 226 (0.25) |
| **Years of diagnosis** | |
| 2004–2008 | 23033 (25.84) |
| 2009–2013 | 28283 (31.73) |
| 2014–2018 | 37831 (42.44) |
| **Histologic type** | |
| LADC | 50863 (57.06) |
| LSCC | 21215 (23.80) |
| Others | 17069 (19.15) |
| **Location** | |
| Upper lobe | 53775 (60.32) |
| Middle lobe | 4966 (5.57) |
| Lower lobe | 28808 (32.32) |
| Unknown | 1598(1.79) |
| **Surgery** | |
| Local[a] | 581 (0.65) |
| Wedge | 13287 (14.90) |
| Segmental | 3389 (3.80) |
| Lobe | 39112 (43.87) |
| NOS | 2215 (2.48) |
| No | 30563 (34.28) |
| **Chemotherapy** | |
| Yes | 5068 (5.68) |
| No/unknown | 84079 (94.32) |
| **Radiotherapy** | |
| Yes | 22968 (25.76) |
| No/unknown | 66179 (74.24) |
| **Marital status** | |
| Married[b] | 46448 (52.10) |
| Single[c] | 38369 (43.04) |
| Unknown | 4330 (4.86) |
| **Treatment modality** | |
| Observation | 8055 (9.04) |
| CA | 1058 (1.19) |
| RA | 19688 (22.08) |

(*Continued*)

**Table 1.** (Continued)

| Variable | Case (%) |
|---|---|
| RC | 1762 (1.98) |
| SA | 55262 (61.99) |
| SC | 1804 (2.02) |
| SR | 1074 (1.20) |
| SRC | 444 (0.50) |

**Abbreviations:** CA: Chemotherapy alone; LADC: Lung adenocarcinoma; LSCC: Lung squamous cell cancer; NOS: Not otherwise specified; RA: Radiation alone; RC: Radiation+Chemotherapy; SA: Surgery alone; SC: Surgery +Chemotherapy; SR: Surgery+Radiation; SRC: Surgery+Radiation+Chemotherapy.

[a] Local tumor destruction (includes laser ablation, cryosurgery, electrocautery and fulguration).

[b] Including marital status: Married or with partner.

[c] Including marital status: Single, divorced/separated or widowed.

(**S7B and S7D Fig** in S1 File). In the LADC subgroup, the order of modalities based on an increasing HR was as follows: SA, SC, SR, RA, SRC, observation, RC, and CA (**S8B and S8D Fig** in S1 File). In the diagnosed 2004–2008 subgroup, the order of modalities based on an increasing HR was as follows: SA, SC, SR, SRC, RA, RC, observation, and CA (**S9B and S9D Fig** in S1 File). Except for the subgroups mentioned above, the order of modalities based on an increasing HR was the same as the overall trend (**S1B and S1D Fig** in S1 File**; S10-S17B and S17D Figs** in S1 File).

In the analysis of treatment modalities, the SA group had the best OS (sequence of treatment efficacy: SA, SC, SR, SRC, RA, RC, CA, and observation) and LCSS, and similar results were found in most subgroup analyses (P<0.001). In the LSCC subgroup, SC was associated with the best OS (HR: 0.20, 95% CI: 0.17–0.23 P<0.001), but SC and SA were not significantly different (HR: 0.94, 95% CI: 0.82–1.08 P = 0.39). The optimal therapeutic modality sequence was as follows: SA, SC, SR, RA, SRC, RC, observation, and CA. Thus, SA was the optimal treatment modality for stage IA NSCLC.

## Comparison of OS and LCSS among different surgical modalities

To determine the optimal surgical modality, patients who received SA were divided into the following groups: local tumor destruction (497, 0.90%), wedge resection (12128, 21.95%), segmentectomy (3194, 5.78%), lobectomy (37511, 67.88%) and NOS (1932, 3.50%). The baseline characteristics of stage IA NSCLC patients treated by different surgical modalities are reported in **Table 5**. Subgroup analyses were not performed due to the low inclusion of American Indian/Alaska Native individuals.

Comparisons of OS and LCSS among different surgical modalities in patients in the entire cohort were carried out. Wedge resection was associated with a longer OS (HR: 0.52, 95% CI: 0.46–0.58, P<0.0001) and LCSS (HR: 0.50, 95% CI: 0.43–0.59, P<0.0001) than local tumor destruction; segmentectomy was associated with a longer OS (HR: 0.80, 95% CI: 0.75–0.85, P<0.0001) and LCSS (HR: 0.80, 95% CI: 0.73–0.88, P<0.0001) than wedge resection; and lobectomy was associated with a longer OS (HR: 0.79, 95% CI: 0.74–0.84, P<0.0001) and LCSS (HR: 0.76, 95% CI: 0.70–0.83, P<0.0001) than segmentectomy (**Fig 4**).

The overall HR for OS of the entire cohort among different surgical modalities increased as follows: lobectomy, segmentectomy, wedge resection, and local tumor destruction (**Fig 5A and 5C**). Similar results were found in all subgroup analyses (**S18-S35A and S35C Figs** in S1 File).

**Table 2. Characteristics of patients with stage IA NSCLC treated by multiple treatment modalities.**

| Variables | Observation | CA | RA | RC | SA | SC | SR | SRC | Overall Cohort N = 89147 |
|---|---|---|---|---|---|---|---|---|---|
| | N (%) | N (%) | N (%) | N (%) | N (%) | N (%) | N (%) | N (%) | N (%) |
| **Age** | | | | | | | | | |
| ≤65 | 1363 (16.92) | 251 (23.72) | 2572 (13.06) | 418 (23.72) | 17495 (31.66) | 787 (43.63) | 261 (24.30) | 183 (41.22) | 23330 (26.17) |
| >65 | 6692 (83.08) | 807 (76.28) | 17116 (86.94) | 1344 (76.28) | 37767 (68.34) | 1017 (56.37) | 813 (75.70) | 261 (58.78) | 65817 (73.83) |
| **Sex** | | | | | | | | | |
| Female | 4229 (52.50) | 552 (52.17) | 10564 (53.66) | 865 (49.09) | 31581 (57.15) | 1035 (57.37) | 562 (52.33) | 236 (53.15) | 49624 (55.67) |
| Male | 3826 (47.50) | 506 (47.83) | 9124 (46.34) | 897 (50.91) | 23681 (42.85) | 769 (42.63) | 512 (47.67) | 208 (46.85) | 39523 (44.33) |
| **Ethnicity** | | | | | | | | | |
| White | 6541 (81.20) | 849 (80.25) | 17028 (86.49) | 1493 (84.73) | 47280 (85.56) | 1550 (85.92) | 930 (86.59) | 385 (86.71) | 76056 (85.32) |
| Black | 1002 (12.44) | 125 (11.81) | 1805 (9.17) | 199 (11.29) | 4313 (7.80) | 145 (8.04) | 110 (10.24) | 40 (9.01) | 7739 (8.68) |
| Asian or Pacific Islander | 450 (5.59) | 80 (7.56) | 694 (3.52) | 62 (3.52) | 3287 (5.95) | 98 (5.43) | 32 (2.98) | 18 (4.05) | 4721 (5.30) |
| American Indian/Alaska Native | 39 (0.48) | 3 (0.28) | 129 (0.66) | 7 (0.40) | 219 (0.40) | 5 (0.28) | 2 (0.19) | 1 (0.23) | 405 (0.45) |
| Unknown | 23 (0.29) | 1 (0.09) | 32 (0.16) | 1 (0.06) | 163 (0.29) | 6 (0.33) | 0 (0.00) | 0 (0.00) | 226 (0.25) |
| **Years of diagnosis** | | | | | | | | | |
| 2004–2008 | 2187 (27.15) | 339 (32.04) | 2188 (11.11) | 593 (33.65) | 16296 (29.49) | 855 (47.39) | 378 (35.20) | 197 (44.37) | 23033 (25.84) |
| 2009–2013 | 2554 (31.71) | 365 (34.50) | 5890 (29.92) | 543 (30.82) | 17890 (32.37) | 494 (27.38) | 395 (36.78) | 152 (34.23) | 28283 (31.73) |
| 2014–2018 | 3314 (41.14) | 354 (33.46) | 11610 (58.97) | 626 (35.53) | 21076 (38.14) | 455 (25.22) | 301 (28.03) | 95 (21.40) | 37831 (42.44) |
| **Histologic type** | | | | | | | | | |
| LADC | 3721 (46.19) | 565 (53.40) | 9314 (47.31) | 785 (44.55) | 34574 (62.56) | 1110 (61.53) | 553 (51.49) | 241 (54.28) | 50863 (57.06) |
| LSCC | 1765 (21.91) | 264 (24.95) | 5868 (29.80) | 585 (33.20) | 11945 (21.62) | 333 (18.46) | 338 (31.47) | 117 (26.35) | 21215 (23.80) |
| Others | 2569 (31.89) | 229 (21.64) | 4506 (22.89) | 392 (22.25) | 8743 (15.82) | 361 (20.01) | 183 (17.04) | 86 (19.37) | 17069 (19.15) |
| **Location** | | | | | | | | | |
| Upper lobe | 4725 (58.66) | 586 (55.39) | 11971 (60.80) | 1100 (62.43) | 33385 (60.41) | 1093 (60.59) | 644 (59.96) | 271 (61.04) | 53775 (60.32) |
| Middle lobe | 439 (5.45) | 64 (6.05) | 933 (4.74) | 78 (4.43) | 3276 (5.93) | 97 (5.38) | 50 (4.66) | 29 (6.53) | 4966 (5.57) |
| Lower lobe | 2558 (31.76) | 365 (34.50) | 6423 (32.62) | 510 (28.94) | 17888 (32.37) | 577 (31.98) | 356 (33.15) | 131 (29.50) | 28808 (32.32) |
| Unknown | 333 (4.13) | 43 (4.06) | 361 (1.83) | 74 (4.20) | 713 (1.29) | 37 (2.05) | 24 (2.23) | 13 (2.93) | 1598(1.79) |
| **Marital status** | | | | | | | | | |
| Married[a] | 3135 (38.92) | 520 (49.15) | 8944 (45.43) | 911 (51.70) | 31054 (56.19) | 1062 (58.87) | 559 (52.05) | 263 (59.23) | 46448 (52.10) |
| Single[b] | 4467 (55.46) | 481 (45.46) | 9728 (49.41) | 786 (44.61) | 21610 (39.10) | 663 (36.75) | 466 (43.39) | 168 (37.84) | 38369 (43.04) |
| Unknown[b] | 453 (5.62) | 57 (5.39) | 1016 (5.16) | 65 (3.69) | 2598 (4.70) | 79 (4.38) | 49 (4.56) | 13 (2.93) | 4330 (4.86) |

**Abbreviations:** CA: Chemotherapy alone; LADC: Lung adenocarcinoma; LSCC: Lung squamous cell cancer; RA: Radiation alone; RC: Radiation+Chemotherapy; SA: Surgery alone; SC: Surgery+Chemotherapy; SR: Surgery+Radiation; SRC: Surgery+Radiation+Chemotherapy.

[a] Including marital status: Married or with partner.

[b] Including marital status: Single, divorced/separated or widowed.

**Table 3. Univariate and Multivariate Cox regression analysis for overall survival (OS) and lung cancer-specific survival (LCSS) in patients with IA NSCLC treated by multiple treatment modalities.**

| Variable | OS | | | | LCSS | | | |
|---|---|---|---|---|---|---|---|---|
| | Univariate | | Multivariate | | Univariate | | Multivariate | |
| | HR (95%CI) | P-value | HR (95%CI) | P-value | HR (95%CI) | P-value | HR (95% CI) | P-value |
| **Age** | | | | | | | | |
| ≤65 | 1.00 (Ref) | | 1.00 (Ref) | | 1.00 (Ref) | | 1.00 (Ref) | |
| >65 | 1.96(1.91–2.01) | <0.001 | 1.63(1.59–1.67) | <0.001 | 1.67(1.62–1.72) | <0.001 | 1.39 (1.34–1.43) | <0.001 |
| **Sex** | | | | | | | | |
| Female | 1.00 (Ref) | | 1.00 (Ref) | | 1.00 (Ref) | | 1.00 (Ref) | |
| Male | 1.42(1.39–1.45) | <0.001 | 1.41(1.39–1.44) | <0.001 | 1.36(1.32–1.39) | <0.001 | 1.35 (1.31–1.38) | <0.001 |
| **Ethnicity** | | | | | | | | |
| White | 1.00 (Ref) | | 1.00 (Ref) | | 1.00 (Ref) | | 1.00 (Ref) | |
| Black | 1.06(1.02–1.10) | 0.001 | 0.98(0.95–1.01) | 0.247 | 1.10(1.05–1.15) | <0.001 | 0.99 (0.95–1.04) | 0.709 |
| Asian or Pacific Islander | 0.64(0.61–0.68) | <0.001 | 0.72(0.69–0.76) | <0.001 | 0.68(0.63–0.73) | <0.001 | 0.76 (0.71–0.82) | <0.001 |
| American Indian/Alaska Native | 1.11(0.97–1.28) | 0.132 | 1.07(0.94–1.24) | 0.309 | 1.14(0.94–1.38) | 0.179 | 1.09 (0.91–1.32) | 0.349 |
| Unknown | 0.19(0.12–0.31) | <0.001 | 0.23(0.14–0.36) | <0.001 | 0.13(0.06–0.29) | <0.001 | 0.16 (0.07–0.35) | <0.001 |
| **Years of diagnosis** | | | | | | | | |
| 2004–2008 | 1.00 (Ref) | | 1.00 (Ref) | | 1.00 (Ref) | | 1.00 (Ref) | |
| 2009–2013 | 0.97(0.95–0.99) | 0.002 | 0.88(0.86–0.90) | <0.001 | 0.89(0.86–0.91) | <0.001 | 0.82 (0.80–0.85) | <0.001 |
| 2014–2018 | 0.83(0.80–0.85) | <0.001 | 0.70(0.68–0.72) | <0.001 | 0.71(0.68–0.73) | <0.001 | 0.61 (0.59–0.64) | <0.001 |
| **Histologic type** | | | | | | | | |
| LADC | 1.00 (Ref) | | 1.00 (Ref) | | 1.00 (Ref) | | 1.00 (Ref) | |
| LSCC | 1.74(0.1.70–1.78) | <0.001 | 1.45(1.42–1.48) | <0.001 | 1.65(1.60–1.70) | <0.001 | 1.39 (1.35–1.43) | <0.001 |
| Others | 1.62(1.58–1.66) | <0.001 | 1.23(1.20–1.26) | <0.001 | 1.59(1.54–1.65) | <0.001 | 1.19 (1.15–1.23) | <0.001 |
| **Location** | | | | | | | | |
| Upper lobe | 1.00 (Ref) | | 1.00 (Ref) | | 1.00 (Ref) | | | |
| Middle lobe | 0.92(0.88–0.96) | <0.001 | 0.99(0.95–1.03) | 0.649 | 0.95(0.90–1.02) | 0.153 | | |
| Lower lobe | 0.99(0.97–1.01) | 0.466 | 1.00(0.98–1.02) | 0.714 | 0.99(0.97–1.02) | 0.670 | | |

*(Continued)*

**Table 3.** (Continued)

| Variable | OS | | | | LCSS | | | |
| --- | --- | --- | --- | --- | --- | --- | --- | --- |
| | Univariate | | Multivariate | | Univariate | | Multivariate | |
| | HR (95%CI) | P-value | HR (95%CI) | P-value | HR (95%CI) | P-value | HR (95% CI) | P-value |
| Unknown | 1.40(1.31–1.49) | <0.001 | 1.12(1.05–1.19) | 0.001 | 1.55(1.42–1.69) | <0.001 | | |
| **Marital status** | | | | | | | | |
| Married[a] | 1.00 (Ref) | | 1.00 (Ref) | | 1.00 (Ref) | | 1.00 (Ref) | |
| Single[b] | 1.29(1.26–1.31) | <0.001 | 1.24(1.21–1.26) | <0.001 | 1.29(1.26–1.33) | <0.001 | 1.21 (1.18–1.25) | <0.001 |
| Unknown | 1.11(1.06–1.17) | <0.001 | 1.08(1.03–1.14) | 0.001 | 1.03(0.96–1.1) | 0.452 | 1.00 (0.93–1.07) | 0.997 |
| **Treatment modality** | | | | | | | | |
| Observation | 1.00 (Ref) | | 1.00 (Ref) | | 1.00 (Ref) | | 1.00 (Ref) | |
| CA | 0.91(0.85–0.98) | 0.012 | 0.94(0.88–1.01) | 0.109 | 1.11(1.02–1.21) | 0.016 | 1.13 (1.04–1.24) | 0.005 |
| RA | 0.56(0.54–0.58) | <0.001 | 0.57(0.55–0.58) | <0.001 | 0.49(0.47–0.51) | <0.001 | 0.51 (0.49–0.53) | <0.001 |
| RC | 0.72(0.68–0.76) | <0.001 | 0.70(0.66–0.74) | <0.001 | 0.87(0.81–0.94) | <0.001 | 0.84 (0.78–0.90) | <0.001 |
| SA | 0.20(0.20–0.21) | <0.001 | 0.23(0.22–0.23) | <0.001 | 0.18(0.17–0.19) | <0.001 | 0.20 (0.19–0.20) | <0.001 |
| SC | 0.25(0.24–0.27) | <0.001 | 0.29(0.27–0.31) | <0.001 | 0.32(0.30–0.35) | <0.001 | 0.35 (0.32–0.38) | <0.001 |
| SR | 0.42(0.39–0.45) | <0.001 | 0.41(0.38–0.45) | <0.001 | 0.45(0.41–0.50) | <0.001 | 0.44 (0.40–0.49) | <0.001 |
| SRC | 0.46(0.41–0.52) | <0.001 | 0.51(0.45–0.57) | <0.001 | 0.59(0.52–0.68) | <0.001 | 0.62 (0.54–0.70) | <0.001 |

**Abbreviations:** CA: Chemotherapy alone; CI: Confidence interval; HR: Hazard ratio; OS: Overall survival; LADC: Lung adenocarcinoma; LCSS: Lung cancer-specific survival; LSCC: Lung squamous cell cancer; RA: Radiation alone; RC: Radiation+Chemotherapy; SA: Surgery alone; SC: Surgery+Chemotherapy; SR: Surgery+Radiation; SRC: Surgery+Radiation+Chemotherapy.

[a] Including marital status: Married or with partner.

[b] Including marital status: Single, divorced/separated or widowed.

The overall HR for LCSS of the entire cohort among different surgical modalities increased as follows: lobectomy, segmentectomy, wedge resection and local tumor destruction (**Fig 5B and 5D**). Similar results were found in all subgroup analyses (**S18-S35B and S35D Figs** in S1 File).

In the analysis of different surgical modalities, lobectomy was associated with the best OS and LCSS (sequence of treatment efficacy: lobectomy, segmentectomy, wedge resection, and local tumor destruction), and similar results were found in all subgroup analyses (P<0.001). The optimal surgical modality sequence was as follows: lobectomy, segmentectomy, wedge resection, and local tumor destruction. Lobectomy was associated with the best prognosis.

**Table 4. Univariate and Multivariate Cox regression analysis for overall survival (OS) and lung cancer-specific survival (LCSS) in patients with IA NSCLC treated by surgery alone.**

| Variable | OS | | | | LCSS | | | |
|---|---|---|---|---|---|---|---|---|
| | Univariate | | Multivariate | | Univariate | | Multivariate | |
| | HR (95%CI) | P-value | HR (95%CI) | P-value | HR (95%CI) | P-value | HR (95%CI) | P-value |
| **Age** | | | | | | | | |
| ≤65 | 1.00 (Ref) | | 1.00 (Ref) | | 1.00 (Ref) | | 1.00 (Ref) | |
| >65 | 1.91(1.85–1.97) | <0.001 | 1.78(1.72–1.84) | <0.001 | 1.57(1.50–1.64) | <0.001 | 1.47(1.40–1.54) | <0.001 |
| **Sex** | | | | | | | | |
| Female | 1.00 (Ref) | | 1.00 (Ref) | | 1.00 (Ref) | | 1.00 (Ref) | |
| Male | 1.49(1.45–1.53) | <0.001 | 1.50(1.46–1.54) | <0.001 | 1.42(1.36–1.47) | <0.001 | 1.42(1.37–1.48) | <0.001 |
| **Ethnicity** | | | | | | | | |
| White | 1.00 (Ref) | | 1.00 (Ref) | | 1.00 (Ref) | | 1.00 (Ref) | |
| Black | 0.96(0.91–1.01) | 0.116 | 1.00(0.95–1.05) | 0.852 | 1.01(0.94–1.08) | 0.822 | 1.03(0.96–1.11) | 0.412 |
| Asian or Pacific Islander | 0.62(0.58–0.66) | <0.001 | 0.70(0.65–0.75) | <0.001 | 0.64(0.58–0.71) | <0.001 | 0.72(0.65–0.80) | <0.001 |
| American Indian/ Alaska Native | 0.95(0.76–1.18) | 0.627 | 1.06(0.85–1.32) | 0.593 | 0.88(0.63–1.22) | 0.441 | 0.96(0.69–1.33) | 0.789 |
| Unknown | 0.10(0.04–0.25) | <0.001 | 0.13(0.05–0.31) | <0.001 | 0.04(0.01–0.30) | 0.002 | 0.05(0.01–0.38) | 0.003 |
| **Years of diagnosis** | | | | | | | | |
| 2004–2008 | 1.00 (Ref) | | 1.00 (Ref) | | 1.00 (Ref) | | 1.00 (Ref) | |
| 2009–2013 | 0.86(0.83–0.89) | <0.001 | 0.86(0.84–0.89) | <0.001 | 0.79(0.76–0.83) | <0.001 | 0.80(0.77–0.84) | <0.001 |
| 2014–2018 | 0.61(0.59–0.64) | <0.001 | 0.63(0.60–0.66) | <0.001 | 0.54(0.50–0.57) | <0.001 | 0.55(0.51–0.58) | <0.001 |
| **Histologic type** | | | | | | | | |
| LADC | 1.00 (Ref) | | 1.00 (Ref) | | 1.00 (Ref) | | 1.00 (Ref) | |
| LSCC | 1.78(1.73–1.83) | <0.001 | 1.53(1.49–1.58) | <0.001 | 1.62(1.55–1.69) | <0.001 | 1.42(1.36–1.49) | <0.001 |
| Others | 1.25(1.20–1.30) | <0.001 | 1.22(1.17–1.26) | <0.001 | 1.30(1.23–1.37) | <0.001 | 1.27(1.20–1.34) | <0.001 |
| **Location** | | | | | | | | |
| Upper lobe | 1.00 (Ref) | | 1.00 (Ref) | | 1.00 (Ref) | | | |
| Middle lobe | 0.94(0.88–1.00) | 0.034 | 1.01(0.95–1.07) | 0.789 | 0.98(0.90–1.06) | 0.644 | | |
| Lower lobe | 0.99(0.96–1.02) | 0.416 | 0.99(0.96–1.02) | 0.583 | 1.00(0.96–1.04) | 0.860 | | |
| Unknown | 1.12(1.00–1.26) | 0.055 | 1.11(0.99–1.24) | 0.08 | 1.18(1.01–1.39) | 0.042 | | |
| **Marital status** | | | | | | | | |
| Married[a] | 1.00 (Ref) | | 1.00 (Ref) | | 1.00 (Ref) | | 1.00 (Ref) | |
| Single[b] | 1.22(1.18–1.25) | <0.001 | 1.29(1.25–1.33) | <0.001 | 1.20(1.15–1.25) | <0.001 | 1.25(1.20–1.31) | <0.001 |
| Unknown | 1.03(0.96–1.11) | 0.357 | 1.12(1.04–1.20) | 0.002 | 0.92(0.83–1.02) | 0.12 | 0.99(0.89–1.10) | 0.898 |
| **Surgery modality** | | | | | | | | |
| Local[c] | 1.00 (Ref) | | 1.00 (Ref) | | 1.00 (Ref) | | 1.00 (Ref) | |
| Wedge | 0.52(0.46–0.58) | <0.001 | 0.59(0.53–0.66) | <0.001 | 0.50(0.43–0.59) | <0.001 | 0.58(0.50–0.67) | <0.001 |
| Segmental | 0.41(0.37–0.46) | <0.001 | 0.49(0.43–0.55) | <0.001 | 0.40(0.34–0.48) | <0.001 | 0.48(0.40–0.56) | <0.001 |
| Lobe | 0.32(0.29–0.36) | <0.001 | 0.39(0.35–0.43) | <0.001 | 0.31(0.27–0.36) | <0.001 | 0.36(0.31–0.42) | <0.001 |

**Abbreviations:** CI: Confidence interval; HR: Hazard ratio; OS: Overall survival; LADC: Lung adenocarcinoma; LCSS: Lung cancer-specific survival; LSCC: Lung squamous cell cancer.

[a] Including marital status: Married or with partner.

[b] Including marital status: Single, divorced/separated or widowed.

[c] Local tumor destruction (includes laser ablation, cryosurgery, electrocautery and fulguration).

## Discussion

Currently, LC is the most common cause of cancer-related morbidity and mortality globally [1]. Early diagnosis and therapy are two essential steps in LC treatment [16]. Over the last half century, a deep understanding of the early detection, diagnosis and treatment of NSCLC has developed [17]. Moreover, it is clear that low-dose lung computed tomography can increase

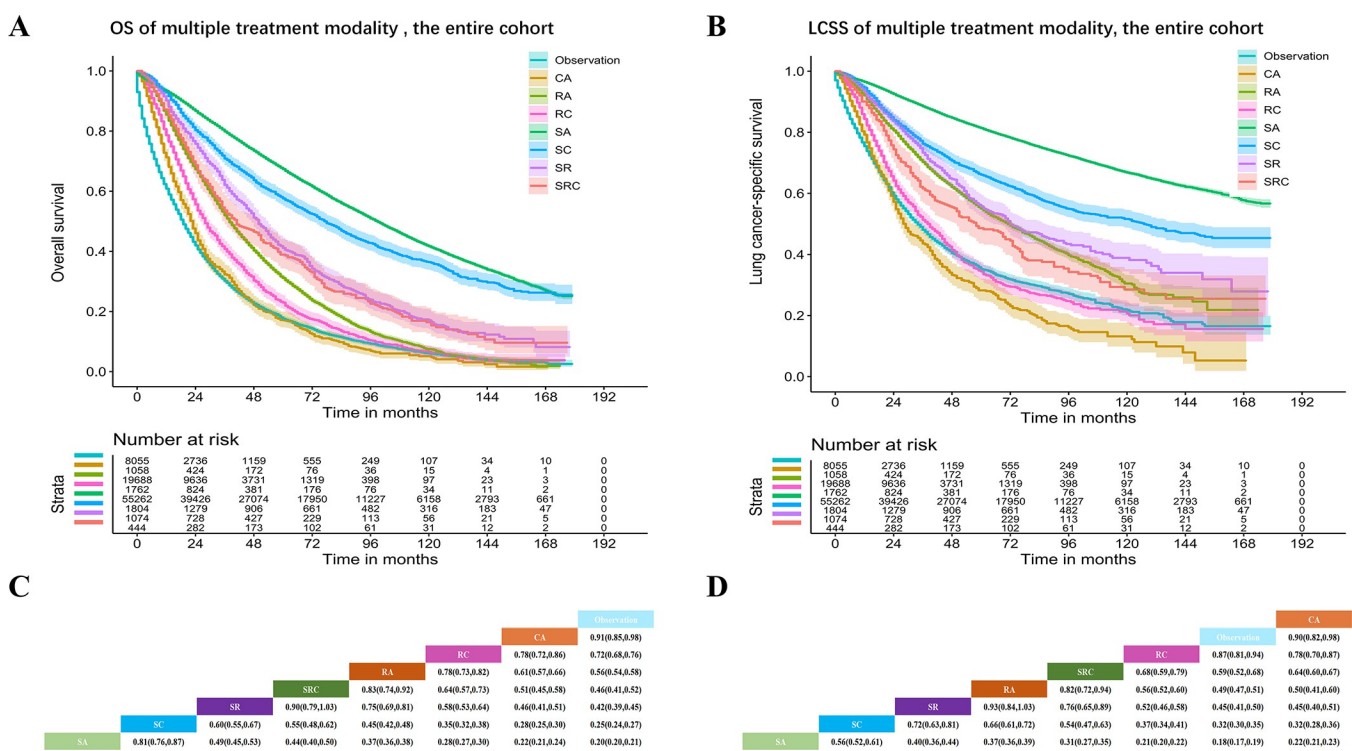

**Fig 2.** Kaplan–Meier curves of OS (**A**) and LCSS (**B**) and HR (95% CI) of OS (**C**) and LCSS (**D**) among multiple treatment modalities in patients with entire cohort.

the detection rate of stage I LC in high-risk populations [18]. An increasing number of patients with early-stage LC are being treated. There are several treatment modalities for stage IA NSCLC, but the best treatment modality for stage IA NSCLC patients remains debatable; therefore, a systematic evaluation of the impact of various treatment methods on patient survival and identification of the optimal treatment modality are needed. This study assessed the survival outcomes associated with different treatment modalities and the effects of different treatment modalities on OS in stage IA NSCLC patients. The novelty of the study is that patients who received observation, radiation, chemotherapy, surgery, and combined treatment (including SR, SC, RC, and SRC) were included. The design of our study may help to determine the best treatment modality for this cohort and determine the sequence of treatment modality options.

Regarding the choice of multiple treatment modalities, according to the results of previous studies, surgery has better survival results than nonsurgery in early-stage NSCLC patients. Primary radiation therapy may be performed for patients who cannot undergo surgery. One study reported no significant effect of neoadjuvant or adjuvant radiation therapy in stage I NSCLC patients [19]. Another study showed that adjuvant radiotherapy achieved good prognosis in local control in patients after surgery. The overall survival rate showed a promising trend [20]. Few studies have included stage IA NSCLC patients in randomized controlled trials of adjuvant therapy thus far. Evidence from a randomized controlled trial showed that adjuvant chemotherapy has no survival benefit in stage IA patients. Adjuvant chemotherapy is not recommended as a therapeutic option for stage IA NSCLC [21]. In our study, the order of treatment modalities based on the HR for OS for the entire cohort revealed the following results: SA (HR: 0.20), SC (HR: 0.25), SR (HR: 0.42), SRC (HR: 0.46), RA (HR: 0.56), RC (HR:

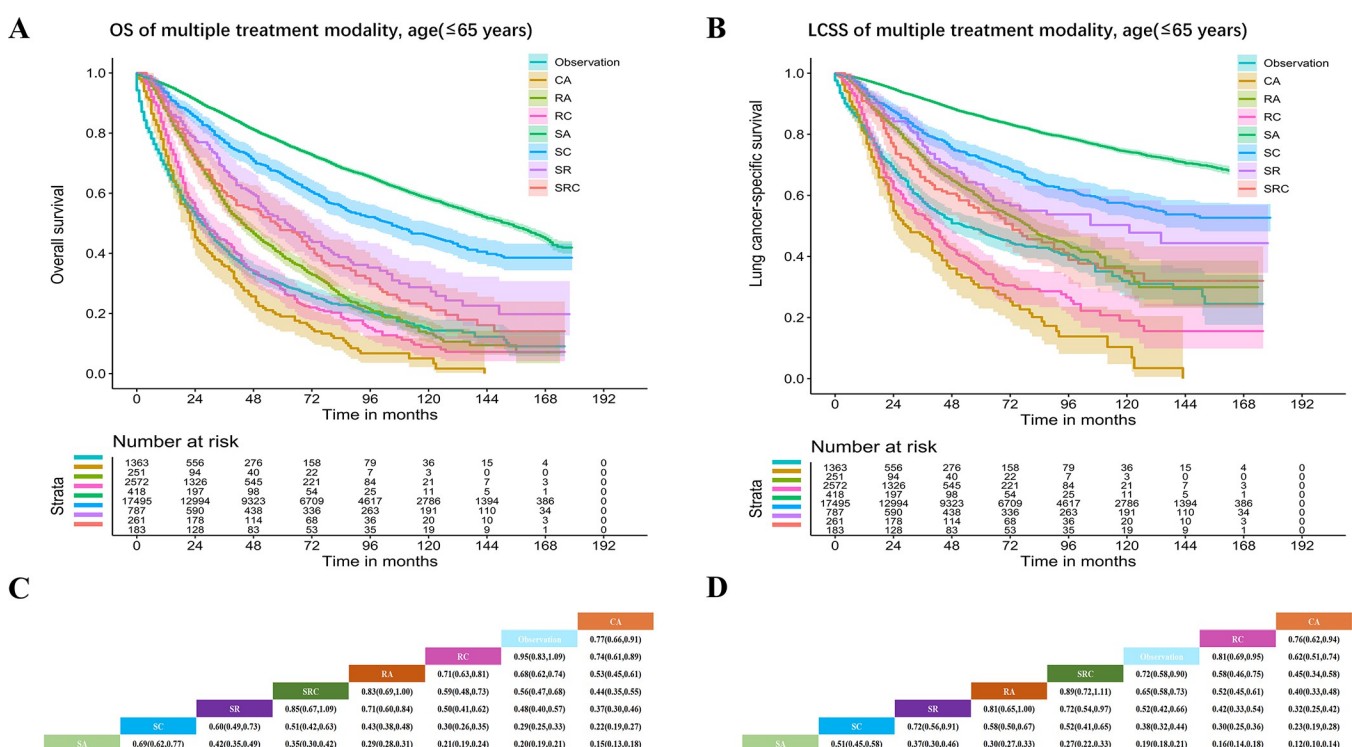

**Fig 3.** Kaplan–Meier curves of OS (**A**) and LCSS (**B**) and HR (95% CI) of OS (**C**) and LCSS (**D**) among multiple treatment modalities in patients with age (≤65 years).

0.72), CA (HR: 0.91) (P<0.001), and observation (HR: Ref). Similar results were found in most subgroup analyses. Our results showed that SA was superior to SC, SR and SRC. In conclusion, adjuvant chemotherapy or adjuvant radiotherapy is unfavorable for the survival of stage IA NSCLC patients. In addition, for stage IA NSCLC patients with inoperable disease, we recommend prioritizing primary radiotherapy; for patients who cannot tolerate primary radiotherapy, chemotherapy should be considered. For stage IA NSCLC patients, SA remains the treatment of choice if the patient can tolerate an operation.

Regarding the choice of surgical modality, research indicates that prognosis following lobectomy is better than that after sublobectomy for most operable IA NSCLC patients [22–24]. However, some studies have shown that segmentectomy has a similar prognosis when compared to lobectomy in stage IA NSCLC patients [25]. The same result was confirmed in the study by Bo et al, who found that lobectomy has no obvious advantage over segmentectomy in elderly patients [26]. Moreover, for nearly two decades, wedge resection has been considered to have a worse prognosis than segmentectomy as surgical technology has developed. Several recently published studies have shown that wedge resection and segmentectomy are equally effective in early-stage NSCLC patients [27]. In recent years, for early-stage NSCLC patients with surgical contraindications, radiofrequency ablation, thermal ablation and cryoablation have been shown to be efficient treatments for improving patient prognosis [5,6,28]. Thus, multiple surgical modalities also need to be systematically evaluated to determine the best surgical treatment. Our study included all the above surgical modalities, and we systematically evaluated the impacts of each. Due to the wide variety of other surgical procedures, the small number of patients who underwent this procedure and the low statistical significance, we did not include it in the survival outcome comparisons. In our study, the order of different

**Table 5. Characteristics of patients with stage IA NSCLC treated by surgery alone.**

| Variable | Local* | Wedge | Segmental | Lobe | NOS | Overall Cohort N = 55262 |
|---|---|---|---|---|---|---|
| | N (%) | N (%) | N (%) | N (%) | N (%) | N (%) |
| **Age** | | | | | | |
| ≤65 | 81 (16.30) | 3153 (26.00) | 833 (26.08) | 12714 (33.89) | 714 (36.96) | 17495 (31.66) |
| >65 | 416 (83.70) | 8975 (74.00) | 2361 (73.92) | 24797 (66.11) | 1218 (63.04) | 37767 (68.34) |
| **Sex** | | | | | | |
| Female | 278 (55.94) | 6944 (57.26) | 1936 (60.61) | 21403 (57.06) | 1020 (52.80) | 31581 (57.15) |
| Male | 219 (44.06) | 5184 (42.74) | 1258 (39.39) | 16108 (42.94) | 912 (47.20) | 23681 (42.85) |
| **Ethnicity** | | | | | | |
| White | 443 (89.13) | 10534 (86.86) | 2773 (86.82) | 31905 (85.06) | 1625 (84.11) | 47280 (85.56) |
| Black | 35 (7.04) | 962 (7.93) | 228 (7.14) | 2912 (7.76) | 176 (9.11) | 4313 (7.80) |
| Asian or Pacific Islander | 18 (3.62) | 545 (4.49) | 175 (5.48) | 2425 (6.46) | 124 (6.42) | 3287 (5.95) |
| American Indian/Alaska Native | 1 (0.20) | 48 (0.40) | 8 (0.25) | 158 (0.42) | 4 (0.21) | 219 (0.40) |
| Unknown | 0 (0.00) | 39 (0.32) | 10 (0.31) | 111 (0.30) | 3 (0.16) | 163 (0.29) |
| **Years of diagnosis** | | | | | | |
| 2004–2008 | 209 (42.05) | 3294 (27.16) | 758 (23.73) | 11157 (29.74) | 878 (45.45) | 16296 (29.49) |
| 2009–2013 | 159 (31.99) | 4045 (33.35) | 948 (29.68) | 12185 (32.48) | 553 (28.62) | 17890 (32.37) |
| 2014–2018 | 129 (25.96) | 4789 (39.49) | 1488 (46.59) | 14169 (37.77) | 501 (25.93) | 21076 (38.14) |
| **Histologic type** | | | | | | |
| LADC | 264 (53.12) | 7398 (61.00) | 2008 (62.87) | 23733 (63.27) | 1171 (60.61) | 34574 (62.56) |
| LSCC | 119 (23.94) | 2782 (22.94) | 640 (20.04) | 7937 (21.16) | 467 (24.17) | 11945 (21.62) |
| Others | 114 (22.94) | 1948 (16.06) | 546 (17.09) | 5841 (15.57) | 294 (15.22) | 8743 (15.82) |
| **Location** | | | | | | |
| Upper lobe | 297 (59.76) | 7404 (61.05) | 1822 (57.04) | 22785 (60.74) | 1077 (55.75) | 33385 (60.41) |
| Middle lobe | 28 (5.63) | 562 (4.63) | 60 (1.88) | 2479 (6.61) | 147 (7.61) | 3276 (5.93) |
| Lower lobe | 162 (32.60) | 4035 (33.27) | 1282 (40.14) | 11826 (31.53) | 583 (30.18) | 17888 (32.37) |
| Unknown | 10 (2.01) | 127 (1.05) | 30 (0.94) | 421 (1.12) | 125 (6.47) | 713 (1.29) |
| **Marital status** | | | | | | |
| Married[a] | 221 (44.47) | 6605 (54.46) | 1748 (54.73) | 21439 (57.15) | 1041 (53.88) | 31054 (56.19) |
| Single[b] | 262 (52.72) | 4906 (40.45) | 1284 (40.20) | 14382 (38.34) | 776 (40.17) | 21610 (39.10) |
| Unknown[c] | 14 (2.82) | 617 (5.09) | 162 (5.07) | 1690 (4.51) | 115 (5.95) | 2598 (4.70) |

**Abbreviations:** LADC: Lung adenocarcinoma; LSCC: Lung squamous cell cancer; NOS: Not otherwise specified.

[a] Including marital status: Married or with partner

[b] Including marital status: Single, divorced/separated or widowed.

surgical modalities based on OS for the entire cohort was as follows: lobectomy (HR: 0.32), segmentectomy (HR: 0.41), wedge resection (HR: 0.52) and local tumor destruction (HR: Ref). Similar results were found in all subgroup analyses (P<0.001). Our study results also further corroborate that compared to the other three surgical modalities, lobectomy is associated with the best prognosis.

The study strengths include the following. First, a large sample of patients, whose data were obtained from a large multicenter clinical database, was included. Second, multiple treatments were assessed, and the effect of different treatments on the prognosis of IA NSCLC patients was systematically analyzed. However, there are some limitations in our study. First, this was a retrospective descriptive study; thus, the presence of indication and selection bias is inevitable compared with prospective research, and the study is only descriptive. Second, The SEER database lacks some important information, such as the inclusion criteria for populations with

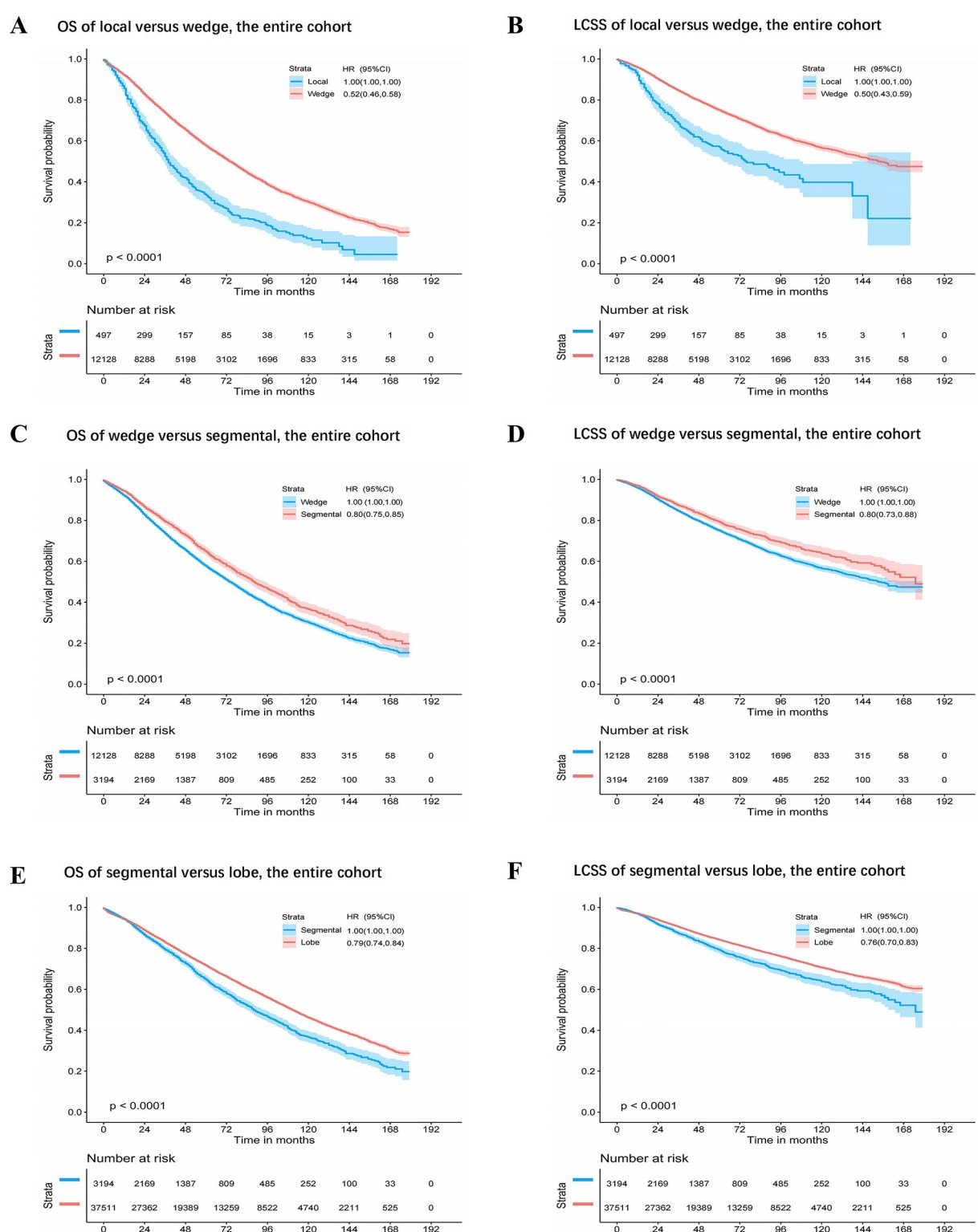

**Fig 4. Kaplan–Meier curves of OS and LCSS among two surgery modalities in patients with entire cohort.** Comparison of OS (**A**) and LCSS (**B**) between local tumor destruction and wedge; comparison of OS (**C**) and LCSS (**D**) between wedge and segmental; and comparison of OS (**E**) and LCSS (**F**) between segmentectomy and lobectomy.

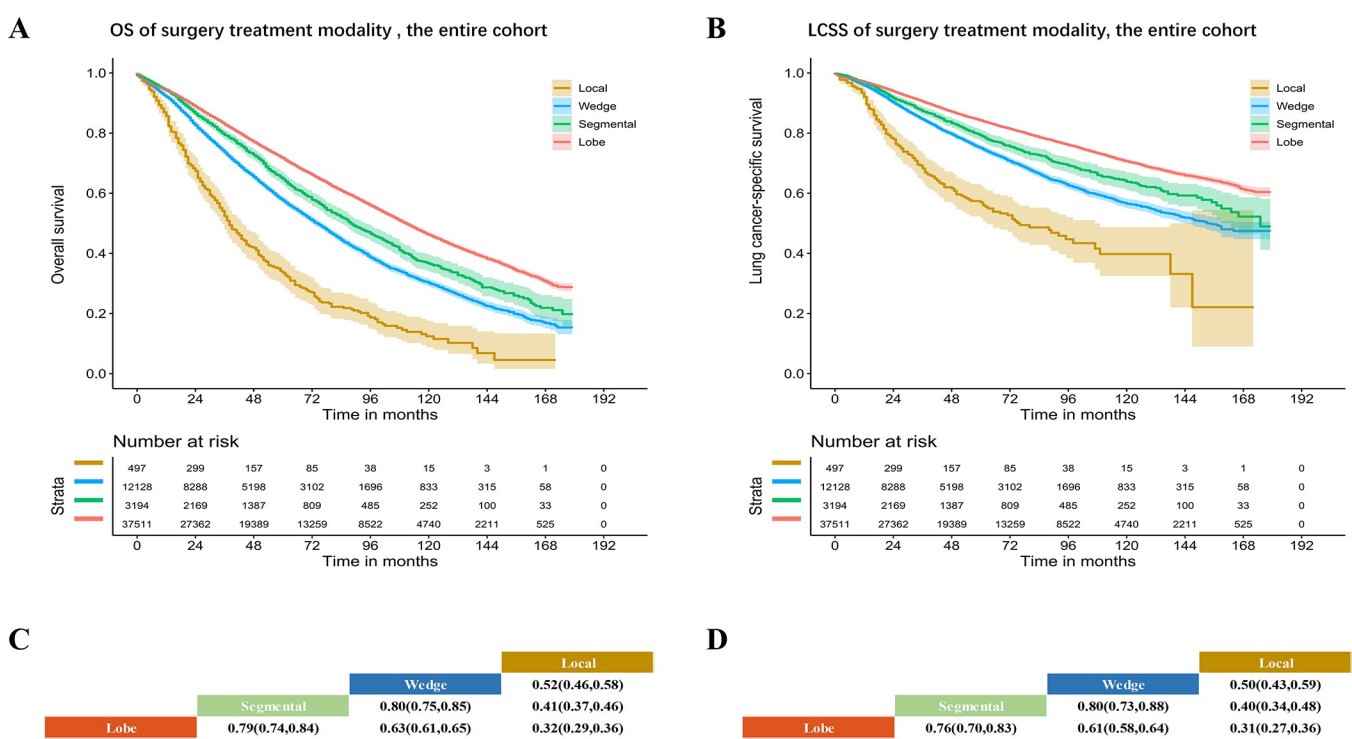

**Fig 5.** Kaplan–Meier curves of OS (**A**) and LCSS (**B**) and HR (95% CI) of OS (**C**) and LCSS (**D**) among different surgery modalities in patients with entire cohort.

different treatment options, status of surgical margins, pathological subtype, tumor size, image feature, etc. Which may have influenced our results. Third, we based the treatment on IA NSCLC patients in the USA, which may not be applicable to patients with IA NSCLC from other countries or of other ethnicities. Due to the above limitations, our findings need to be validated in future prospective studies.

## Conclusions

In summary, SA appeared to be the optimal treatment modality for **patients** with stage IA NSCLC, for which lobectomy was associated with the best prognosis. The sequence of treatment modality options from most effective to least effective was SA, SC, SR, SRC, RA, RC, CA, and observation, and that of surgical treatment modality options was lobectomy, segmentectomy, wedge resection and local tumor destruction. Because of the limitations described above, validation in large prospective studies is needed. Nevertheless, our research results help to identify the optimal therapeutic schedule for these patients.

## Supporting information

**S1 File.**
(PDF)

**S2 File.**
(PDF)

## Acknowledgments

The authors would like to thank all public health workers who gathered these cancer statistics and all patients and doctors relevant to this research.

## Author Contributions

**Conceptualization:** Bo Wu, Xiang Zhang, Nan Feng, Zhuozheng Hu, Jiajun Wu, Weijun Zhou, Yiping Wei, Wenxiong Zhang, Kang Wang.

**Data curation:** Bo Wu, Xiang Zhang, Nan Feng, Zhuozheng Hu, Jiajun Wu, Weijun Zhou, Yiping Wei, Wenxiong Zhang, Kang Wang.

**Formal analysis:** Bo Wu, Xiang Zhang, Nan Feng, Zhuozheng Hu, Jiajun Wu, Weijun Zhou, Yiping Wei, Wenxiong Zhang, Kang Wang.

**Funding acquisition:** Wenxiong Zhang.

**Investigation:** Bo Wu, Yiping Wei, Wenxiong Zhang, Kang Wang.

**Methodology:** Bo Wu, Yiping Wei, Wenxiong Zhang, Kang Wang.

**Project administration:** Bo Wu, Wenxiong Zhang, Kang Wang.

**Resources:** Bo Wu, Wenxiong Zhang, Kang Wang.

**Software:** Bo Wu, Wenxiong Zhang, Kang Wang.

**Supervision:** Bo Wu, Wenxiong Zhang, Kang Wang.

**Validation:** Bo Wu, Wenxiong Zhang, Kang Wang.

**Visualization:** Bo Wu, Wenxiong Zhang, Kang Wang.

**Writing – original draft:** Bo Wu, Xiang Zhang, Nan Feng, Zhuozheng Hu, Jiajun Wu, Weijun Zhou, Yiping Wei, Wenxiong Zhang, Kang Wang.

**Writing – review & editing:** Bo Wu, Wenxiong Zhang, Kang Wang.

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
