## [Decision Letter · Decision Letter 0]

11 Dec 2023

PONE-D-23-35311Treatment strategies for stage IA non-small cell lung cancer: A SEER-based population studyPLOS ONE

Dear Dr. Zhang,

Thank you for submitting your manuscript to PLOS ONE. After careful consideration, we feel that it has merit but does not fully meet PLOS ONE’s publication criteria as it currently stands. Therefore, we invite you to submit a revised version of the manuscript that addresses the points raised during the review process.

We look forward to receiving your revised manuscript.

Kind regards,

Luca Bertolaccini, M.D., Ph.D.

Academic Editor

PLOS ONE

A clean copy of the edited manuscript (uploaded as the new *manuscript* file)”.

Additional Editor Comments:

The reviewers have emphasized issues that require a careful and thorough manuscript revision.

No commitment to publication can be made at this point.

Reviewers' comments:

Reviewer's Responses to Questions

**Comments to the Author**

1. Is the manuscript technically sound, and do the data support the conclusions?

Reviewer #1: Partly

Reviewer #2: Yes

Reviewer #3: Yes

2. Has the statistical analysis been performed appropriately and rigorously? 

Reviewer #1: Yes

Reviewer #2: Yes

Reviewer #3: Yes

3. Have the authors made all data underlying the findings in their manuscript fully available?

Reviewer #1: Yes

Reviewer #2: Yes

Reviewer #3: Yes

4. Is the manuscript presented in an intelligible fashion and written in standard English?

Reviewer #1: No

Reviewer #2: Yes

Reviewer #3: Yes

5. Review Comments to the Author

Reviewer #1: The authors explored the treatment strategies for IA stage NSCLC based on SEER population. In overall, the design of this study is lack of novelty and the findings are lack of clinical practice. Several major concerns existed in this study.

1. A great amount of parameters are missing in the SEER database such as the pathological subtype, tumor size, image feature, etc.

2. "Lobectomy was associated with the best prognosis for stage IA NSCLC patients"? It is not consistent with the clinical practice. In other words, this conclusion is too crude.

Reviewer #2: I would like to express to you my gratitude for the opportunity to provide feedback on your paper, "Treatment Strategies for Stage IA Non-Small Cell Lung Cancer: A SEER-Based Population Study,".

Your paper serves as a comprehensive retrospective observational population-based epidemiological study, delving into the survival outcomes of stage IA NSCLC patients who underwent various treatments. Your findings convincingly highlight surgery as the optimal treatment modality for patients with stage IA NSCLC, with lobectomy demonstrating the most favorable prognosis.

As you are well aware, surgery is commonly acknowledged as the preferred approach in early stage lung cancer potentially offering a curative path, thus improving overall patient outcomes. While the article's topic is undeniably intriguing and relevant, it might not significantly push the boundaries of our existing knowledge in this area. Nevertheless, the exploration of such themes remains valuable, and your study contributes meaningfully to the ongoing discourse.

It is important to acknowledge the inherent limitations of your study, arising from its observational nature and the specific population under consideration. However, the article is pleasant to read, english is easy and it ensures efficient information absorption by readers. Despite not fundamentally advancing our understanding, the insights presented in your article are undoubtedly valuable for further investigation into this topic.

Considering the merits of your work, I recommend accepting the article for publication.

Reviewer #3: Thank you for asking me to review this manuscript.

This retrospective observational study investigates the most effective treatment for stage IA lung cancer.

The topic it’s quite interesting and this is a good quality work.

I don’t have any remarks other than the ones you state from line 73 to 85.

6. PLOS authors have the option to publish the peer review history of their article (what does this mean?). If published, this will include your full peer review and any attached files.

Reviewer #1: No

Reviewer #2: **Yes: **Alessio Campisi

Reviewer #3: **Yes: **Francesco Zaraca

---

## [Author Response · Author response to Decision Letter 0]

17 Jan 2024

Dear Editors and Reviewers:

Thank you for your letter and for the reviewers’ comments concerning our manuscript entitled “Treatment strategies for stage IA non-small cell lung cancer: A SEER-based population study” to PLOS ONE (ID: PONE-D-23-35311). We are very sorry for submitting a revised manuscript so late. Those comments are all valuable and very helpful for revising and improving our paper, as well as the important guiding significance to our researches. Revised portion are marked in red in the paper. The main corrections in the paper and the responds to the reviewer’s comments are as following:

Responds to the reviewer’s comments:

Reviewer #1:

1. Response to comment: The authors explored the treatment strategies for IA stage NSCLC based on SEER population. In overall, the design of this study is lack of novelty and the findings are lack of clinical practice. Several major concerns existed in this study.

Response: The authors all appreciate your sincere comments on the manuscript. There are various therapeutic methods for stage IA (T1N0M0) non-small cell lung cancer (NSCLC), and we did a retrospective descriptive study using data from the US SEER database. This study aims to clarify the influence of different treatment methods on survival prognosis in stage IA non-small cell lung cancer patients and to carry out further subgroup analyses. Ultimately, we concluded that surgery alone appeared to be the optimal treatment modality for patients with stage IA NSCLC; specifically, lobectomy was associated with the best prognosis. Unfortunately, this registration study has some limitations. First, this was a retrospective descriptive study; thus, the design of this study is lack of novelty. Second, our findings are only descriptive, our findings are only descriptive. Multiple treatment modalities for lung cancer have not been validated in clinical practice. However, surgical treatments are now practiced clinically. We hope to further explore the optimal surgical treatment for different stage IA NSCLC populations in future studies. We would try to improve the quality of the intended manuscript considering your instructions and we were sure that the study would benefit from these revisions.

2. Response to comment: A great amount of parameters are missing in the SEER database such as the pathological subtype, tumor size, image feature, etc.

Response and changes: Thank you very much for your comments. We are very sorry that due to the limitations of the SEER database we can't access important information such as the pathological subtype, tumor size, image feature, etc. We would like to include more important clinical information in future prospective studies. The author has explained in detail in the discussion section. The comparison before and after modification is as follows (see details in page 17, line 333 to line 345 of the revised manuscript): 

Add: "Second, The SEER database lacks some important information, such as the inclusion criteria for populations with different treatment options, status of surgical margins, pathological subtype, tumor size, image feature, etc. Which may have influenced our results. Third, we based the treatment on IA NSCLC patients in the USA, which may not be applicable to patients with IA NSCLC from other countries or of other ethnicities. Due to the above limitations, our findings need to be validated in future prospective studies." 

3. Response to comment: Lobectomy was associated with the best prognosis for stage IA NSCLC patients"? It is not consistent with the clinical practice. In other words, this conclusion is too crude.

Response and changes: Thank you for your valuable comments. We couldn't agree with you more. The results of randomized trial of lobectomy versus limited resection for T1 N0 non-small cell lung cancer suggest that compared with lobectomy, limited pulmonary resection does not confer improved perioperative morbidity, mortality, or late postoperative pulmonary function. Because of the higher death rate and locoregional recurrence rate associated with limited resection, lobectomy still must be considered the surgical procedure of choice for patients with peripheral T1 N0 non-small cell lung cancer [1]. The results of segmentectomy versus lobectomy in small-sized peripheral non-small-cell lung cancer (JCOG0802/WJOG4607L) suggest that segmentectomy should be the standard surgical procedure for this population of patients [2]. Our conclusions may be too crude for specific populations. However, due to the shortcomings of the SEER database, we cannot further break down specific populations. The inclusion population in our study was all patients with stage IA NSCLC and the result of this study is only descriptive. I am very sorry to say that these problems mentioned above are the shortcomings of our study. We hope in future studies to explore the optimal surgical treatment in specific populations. The authors have discussed these limitations in the discussion section. The comparison before and after modification is as follows (see details in page 17, line 336 to line 345 of the revised manuscript):

References

1. Ginsberg RJ, Rubinstein LV. Randomized trial of lobectomy versus limited resection for T1 N0 non‐small cell lung cancer. Lung Cancer Study Group[J]. Ann Thorac Surg, 1995, 60(3): 615‐622; discussion 622‐623. DOI:10.1016/0003‐4975(95)00537‐u

2. Nakamura K, Saji H, Nakajima R, et al. A phase Ⅲ randomized trial of lobectomy versus limited resection for small‐sized peripheral non‐small cell lung cancer (JCOG0802/WJOG4607L) [J]. Jpn J Clin Oncol, 2010, 40(3): 271‐274. DOI: 10.1093/jjco/hyp156.

Add: "However, there are some limitations in our study. First, this was a retrospective descriptive study; thus, the presence of indication and selection bias is inevitable compared with prospective research, and the study is only descriptive. Second, The SEER database lacks some important information, such as the inclusion criteria for populations with different treatment options, status of surgical margins, pathological subtype, tumor size, image feature, etc. Which may have influenced our results. Third, we based the treatment on IA NSCLC patients in the USA, which may not be applicable to patients with IA NSCLC from other countries or of other ethnicities. Due to the above limitations, our findings need to be validated in future prospective studies."

Thank you for your good and valuable comments on our manuscript!

Reviewer #2:

1. Response to comment: I would like to express to you my gratitude for the opportunity to provide feedback on your paper, "Treatment Strategies for Stage IA Non-Small Cell Lung Cancer: A SEER-Based Population Study,". Your paper serves as a comprehensive retrospective observational population-based epidemiological study, delving into the survival outcomes of stage IA NSCLC patients who underwent various treatments. Your findings convincingly highlight surgery as the optimal treatment modality for patients with stage IA NSCLC, with lobectomy demonstrating the most favorable prognosis. As you are well aware, surgery is commonly acknowledged as the preferred approach in early stage lung cancer potentially offering a curative path, thus improving overall patient outcomes. While the article's topic is undeniably intriguing and relevant, it might not significantly push the boundaries of our existing knowledge in this area. Nevertheless, the exploration of such themes remains valuable, and your study contributes meaningfully to the ongoing discourse.

Response: The authors all appreciate your sincere comments on the manuscript. This study is a retrospective study aimed to address the optimal treatment modality options for stage IA NSCLC patients by comparing overall survival (OS) and lung cancer-specific survival (LCSS) among different treatment methods based on the Surveillance, Epidemiology, and End Results (SEER) database. And the authors concluded that surgery alone appeared to be the optimal treatment modality for patients with stage IA NSCLC, in which lobectomy was associated with the best prognosis. It is well known that surgery is recognized as the treatment of choice for early stage lung cancer. Our results also systematically confirm that surgery is the treatment of choice for early stage lung cancer. We would try to improve the quality of the intended manuscript considering your instructions and we were sure that the study would benefit from these revisions.

2. Response to comment: It is important to acknowledge the inherent limitations of your study, arising from its observational nature and the specific population under consideration. However, the article is pleasant to read, english is easy and it ensures efficient information absorption by readers. Despite not fundamentally advancing our understanding, the insights presented in your article are undoubtedly valuable for further investigation into this topic.

Considering the merits of your work, I recommend accepting the article for publication.

Response and changes: Thank you for your good and valuable comments and recognition! The authors acknowledge the inherent limitations of this study, arising from its observational nature and the specific population under consideration. The authors would like to do multicenter prospective studies in the future to further confirm the optimal treatment for IA NSCLC patients. The authors have discussed these limitations in the discussion section. The comparison before and after modification is as follows (see details in page 17, line 336 to line 345 of the revised manuscript):

Add: "However, there are some limitations in our study. First, this was a retrospective descriptive study; thus, the presence of indication and selection bias is inevitable compared with prospective research, and the study is only descriptive. Second, The SEER database lacks some important information, such as the inclusion criteria for populations with different treatment options, status of surgical margins, pathological subtype, tumor size, image feature, etc. Which may have influenced our results. Third, we based the treatment on IA NSCLC patients in the USA, which may not be applicable to patients with IA NSCLC from other countries or of other ethnicities. Due to the above limitations, our findings need to be validated in future prospective studies."

Thank you for your good and valuable comments on our manuscript!

Reviewer #3:

1. Response to comment: Thank you for asking me to review this manuscript. This retrospective observational study investigates the most effective treatment for stage IA lung cancer. The topic it’s quite interesting and this is a good quality work.

Response: The authors are very grateful for your sincere opinion and recognition of the manuscript. This study systematically evaluates the impact of various treatment methods on patient survival and identification of the optimal treatment modality based on SEER database. The results of this study found that surgery alone appeared to be the optimal treatment modality for patients with stage IA NSCLC, and lobectomy was associated with the best prognosis. We would try to improve the quality of the intended manuscript considering your instructions and we were sure that the study would benefit from these revisions.

2. Response to comment: I don’t have any remarks other than the ones you state from line 73 to 85.

Response and changes: Thank you for your good and valuable comments. We are very sorry that we did not notice this detail. We apologize for putting the strengths and limitations of this study after the summary section of the article. The author has placed the strengths and limitations of this study in the discussion section. The comparison before and after modification is as follows (see details in page 17, line 333 to line 345 of the revised manuscript):

Deletion: "

Strengths and limitations of this study:

First, a large sample of patients, whose data were obtained from a large multicenter clinical database, was included.

Second, multiple treatments were assessed, and the effect of different treatments on the prognosis of IA NSCLC patients was systematically analyzed.

Third, this was a retrospective descriptive study; thus, the presence of indication and selection bias is inevitable compared with prospective research, and the study is only descriptive.

Fourth, the SEER database does not mention the inclusion criteria for the populations receiving the different treatment regimens and the status of the surgical margins, which may have influenced our results.

Fifth, we based the treatment on IA NSCLC patients in the USA, which may not be applicable to patients with IA NSCLC from other countries or of other ethnicities."

Add: "The study strengths include the following. First, a large sample of patients, whose data were obtained from a large multicenter clinical database, was included. Second, multiple treatments were assessed, and the effect of different treatments on the prognosis of IA NSCLC patients was systematically analyzed. However, there are some limitations in our study. First, this was a retrospective descriptive study; thus, the presence of indication and selection bias is inevitable compared with prospective research, and the study is only descriptive. Second, The SEER database lacks some important information, such as the inclusion criteria for populations with different treatment options, status of surgical margins, pathological subtype, tumor size, image feature, etc. Which may have influenced our results. Third, we based the treatment on IA NSCLC patients in the USA, which may not be applicable to patients with IA NSCLC from other countries or of other ethnicities. Due to the above limitations, our findings need to be validated in future prospective studies."

Thank you for your good and valuable comments on our manuscript!

We tried our best to improve the manuscript and made some changes in the manuscript. These changes will not influence framework of the paper. And here we did not list the changes but marked in red in revised paper.

We appreciate for Editors/Reviewers’ warm work earnestly, and hope that the correction will meet with approval.

Once again, thank you very much for your comments and suggestions.

Corresponding Author: 

Name: Wenxiong Zhang 

E-mail: zwx123dr@126.com.

---

## [Decision Letter · Decision Letter 1]

25 Jan 2024

Treatment strategies for stage IA non-small cell lung cancer: A SEER-based population study

PONE-D-23-35311R1

Dear Dr. Zhang,

We’re pleased to inform you that your manuscript has been judged scientifically suitable for publication and will be formally accepted for publication once it meets all outstanding technical requirements.

Kind regards,

Luca Bertolaccini, M.D., Ph.D.

Academic Editor

PLOS ONE

Additional Editor Comments (optional):

Reviewers' comments:

Reviewer's Responses to Questions

**Comments to the Author**

1. If the authors have adequately addressed your comments raised in a previous round of review and you feel that this manuscript is now acceptable for publication, you may indicate that here to bypass the “Comments to the Author” section, enter your conflict of interest statement in the “Confidential to Editor” section, and submit your "Accept" recommendation.

Reviewer #2: All comments have been addressed

Reviewer #3: All comments have been addressed

2. Is the manuscript technically sound, and do the data support the conclusions?

Reviewer #2: Yes

Reviewer #3: Yes

3. Has the statistical analysis been performed appropriately and rigorously? 

Reviewer #2: Yes

Reviewer #3: Yes

4. Have the authors made all data underlying the findings in their manuscript fully available?

Reviewer #2: Yes

Reviewer #3: Yes

5. Is the manuscript presented in an intelligible fashion and written in standard English?

Reviewer #2: Yes

Reviewer #3: Yes

6. Review Comments to the Author

Reviewer #2: You have done an excellent job incorporating our suggestions. In my opinion, the article is now ready for publication. I am inclined to recommend your paper to our colleagues. Thank you for the privilege of working on this collaborative effort.

Reviewer #3: The authors have appropriately corrected the manuscript, improving its scientific value. In my opinion it can be accepted without further revision.

7. PLOS authors have the option to publish the peer review history of their article (what does this mean?). If published, this will include your full peer review and any attached files.

Reviewer #2: **Yes: **Alessio Campisi

Reviewer #3: No
